# Combinatorial Effects of Soluble, Insoluble, and Organic Extracts from Jerusalem Artichokes on Gut Microbiota in Mice

**DOI:** 10.3390/microorganisms8060954

**Published:** 2020-06-24

**Authors:** Hiroyuki Sasaki, Yijin Lyu, Yuki Nakayama, Fumiaki Nakamura, Aya Watanabe, Hiroki Miyakawa, Yoichi Nakao, Shigenobu Shibata

**Affiliations:** 1Laboratory of Physiology and Pharmacology, School of Advanced Science and Engineering, Waseda University, Shinjuku-ku, Tokyo 162-8480, Japan; hiroyuki-sasaki@asagi.waseda.jp (H.S.); ikin@fuji.waseda.jp (Y.L.); yukibecky-6991@akane.waseda.jp (Y.N.); aya_watanabe7115@suou.waseda.jp (A.W.); hgbbst-hiroki@toki.waseda.jp (H.M.); 2Laboratory of Chemical biology, School of Advanced Science and Engineering, Waseda University, Shinjuku-ku, Tokyo 162-8480, Japan; what-will_be.x2@akane.waseda.jp (F.N.); ayocha@waseda.jp (Y.N.); 3Research Institute for Science and Engineering, Waseda University, 3-4-1 Okubo, Shinjuku-ku, Tokyo 169-8555, Japan

**Keywords:** microbiota, Jerusalem artichoke, inulin, organic-soluble materials

## Abstract

Jerusalem artichokes contain high amounts of inulin, which is a prebiotic that supports digestive health, as well as a variety of insoluble fibers and caffeoylquinic acid. The individual impact of these components on gut microbiota is well known; however, the combinatorial effects are less clear. In this investigation, we fractionated Jerusalem artichokes into three parts (water-soluble extract, insoluble extract, and organic extract) and powdered them. Mice were fed a high-fat diet that included one or more of these extracts for 10 days, and then their cecal pH, cecal short-chain fatty acids (SCFAs), and fecal microbiota were evaluated. The combination of the water-soluble and organic extract decreased cecal pH and increased the concentration of SCFAs and led to dynamic changes in the composition of the gut microbiota. These results demonstrate that both the water-soluble and organic extracts in Jerusalem artichokes are bioactive substances that are capable of changing SCFA production and the composition of gut microbiota. Powdered Jerusalem artichokes, rather than inulin supplements, may be superior for promoting a healthy gut.

## 1. Introduction

There are nearly 40 trillion bacteria residing in the intestines of mammals, which comprise the microbiota. Dysbiosis, or imbalance between good and bad bacterial populations in the gut, has been reported to be involved in the development of various diseases [1]. For example, high concentrations of Firmicutes have been implicated in an increased risk of obesity, whereas an abundance of Bacteroidetes appears to suppress body fat accumulation [2]. In an investigation that transplanted germ-free mice with feces from obese and lean mice, the recipients of the former became obese [3]. Other pathogenic bacteria, such as *Fusobacterium* and *Fusobacterium nucleatum*, have been observed in greater amounts among patients with colorectal cancer compared to healthy counterparts [4,5], and for patients with major depression, unusually high levels of Bacteroidetes, Proteobacteria, and Actinobacteria have been detected [6].

Short-chain fatty acids (SCFAs) are produced from the bacterial fermentation and degradation of resistant starches and dietary fibers [7], and they are known to suppress the growth of pathogenic bacteria by lowering intestinal pH, as well as help to regulate metabolism and the immune system [8]. Of the various SCFAs, acetic acid serves as an energy substrate for the liver, propionic acid participates in gluconeogenesis, and butyric acid promotes the induction of regulatory T-cells in the large intestine [9,10].

The composition of microbiota changes in response to the nutritional components of foods consumed, especially those rich in dietary fiber [11]. Often termed prebiotics, dietary fibers are described as nondigestible food ingredients that beneficially affect the host by selectively stimulating the growth and/or activity of one or a limited number of bacterial species already resident in the colon, and thus attempt to improve host health [12]. Inulin, a prominent water-soluble dietary fiber, has gained considerable attention from researchers due to its ability promote mineral absorption [13,14,15], keep blood sugar stable [16], enhance lipid metabolism [16], treat inflammatory bowel disease [17], and improve the composition of microbiota [18,19].

In recent years, contemporary reports have discovered inulin to be particularly concentrated in Jerusalem artichokes [20,21], and it has generally been recognized as the main component responsible for improvements in microbiota [22,23,24,25]. However, to this point, it is well known that vegetables also contain other functional substances such as insoluble dietary fiber, which boosts intestinal health by absorbing water and increasing stool volume to promote peristalsis and defecation [26,27]. Thus, it is presumable that other factors in Jerusalem artichokes may be contributing to its health-promoting properties. Notably, the insoluble fiber, caffeoylquinic acid, coumarin, lectins, and sesquiterpenes [28,29,30,31] present in Jerusalem have been proven to possess bioactivity, although their direct impact on gut microbiota remains to be elucidated.

In this study, we investigated the effects of Jerusalem artichokes versus pure inulin on intestinal microbiota in mice fed a high-fat diet (HFD), aiming specifically to identify: (1) the effects of Jerusalem artichokes versus inulin on microbiota, (2) the effects of water-soluble versus insoluble dietary fiber on microbiota, and (3) the effects of organic extracts from Jerusalem artichokes on microbiota. Mice were fed this diet because previous research by our team demonstrated that inulin helps to blunt the injurious effects of HFDs [19]. Additionally, based on preliminary data, Jerusalem artichokes exert stronger effects on the microbiota rather than inulin alone, so this vegetable was fractionated into three parts: water-soluble extract (rich in inulin), insoluble extract, and organic extract in order to carry out the aims listed above. We hypothesized that Jerusalem artichokes would change the composition of microbiota due to their high inulin content, but this effect would be weaker than that of pure inulin.

## 2. Materials and Methods 

### 2.1. Animals

Male ICR mice (*n* = 94, 8 weeks of age) were purchased from Tokyo Laboratory Animals (Tokyo, Japan). They were housed in individual cages and kept in a room maintained on a 12 h light/12 h dark cycle at 22 °C, 60% humidity, and a light intensity of 100–150 lux. Light-on time was defined as zeitgeber time 0 (ZT0) and lights-off time was defined as ZT12. The mice were given a HFD consisting of 45% kcal from fat (Diet 12451; Research Diets Inc., New Brunswick, NJ, USA), cellulose (Oriental Yeast Co., Ltd., Tokyo, Japan), inulin (Fuji FF; Fuji Nihon Seito Co., Tokyo, Japan), and one or more extractions of Jerusalem artichoke powder (Yasai Factory Co., Ltd, Osaka, Japan) (see Table 1). Experiments were conducted with permission from the Committee for Animal Experimentation of the School of Science and Engineering at Waseda University (permission # 09A11, 10A11), as well as in accordance with the Law (No. 105) and Notification (No. 6) of the Japanese Government.

### 2.2. Fractioning the Jerusalem Artichoke

Jerusalem artichoke powder was dissolved in water and centrifuged (450× *g*, 1 min, room temperature) to separate the water-soluble layer from the insoluble layer. Each layer was freeze-dried for three days to become powdered. Since there was the possibility that some of the water-soluble and insoluble extracts may not have fully separated into their respective layers following centrifugation, the insoluble extract was washed several times with water using filter bell (#1 paper filter in ADVANTEC), and the water-soluble layer was freeze-dried for three days to become powdered. The fraction extracted with water was deemed the “water-soluble extract”. The fraction extracted with methanol and chloroform (1:1) using a filter bell (#1 paper filter in ADVANTEC) was deemed the “organic extract”. The fraction not extracted with water, methanol, or chloroform was deemed the “insoluble extract”. Each extract was freeze-dried for three days and then powdered (Figure 1).

### 2.3. Determining the Inulin/Fructan Content

The amounts of fructan in each extract were determined by the K-FRUC Fructan Assay Kit (Megazyme, Bray, Ireland). This kit can determine the amount of natural fructan and levan-type fructan present. If the fructan is natural, all molecular weights are included in the measurement. If it includes hydrolytic fructans, the amount of fructan will be underestimated by approximately 20%, because the sugars will be reduced to sugar alcohols.

### 2.4. Experimental Design

For the first aim, mice were fed a HFD with 2.5% cellulose (cellulose group), 2.5% inulin (inulin group), or 2.5% Jerusalem artichoke powder (Jerusalem artichoke group) for 10 days. After the feeding period, mice were sacrificed at ZT4 (i.e., 4 h after the onset of the light phase). Cecal pH was measured, and cecal contents and feces were collected.

For the second aim, mice were fed a HFD with 2.5% cellulose (control group), 2.5% Jerusalem artichoke powder (A group), 2.5% soluble powder (B group), 2.5% insoluble powder (C group), or 1.25% soluble + 1.25% insoluble powder (B + C group) for 10 days. After the feeding period, mice were sacrificed at ZT4. Cecal pH was measured, and cecal contents and feces were collected.

For the third aim, mice were fed a HFD with 2.5% cellulose (control group), 2.5% Jerusalem artichoke powder (A group), 2.5% water-soluble extract (B’ group), 2.5% organic extract (D group), 2.5% insoluble extract (E group), 1.25% water-soluble extract + 1.25% organic extract (B’ + D group), or 1.25% water-soluble extract + 1.25% insoluble extract (B’ + E group) for 10 days. After the feeding period, mice were sacrificed at ZT4. Cecal pH was measured, and cecal contents and feces were collected.

### 2.5. Cecal pH Measurement

Cecal pH was measured by inserting an electrode from a pH meter (pH Spear; Eutech Instruments, Vernon Hills, IL, USA) directly into the cecum.

### 2.6. SCFA Measurement

SCFAs were measured using gas chromatography and a flame ionization detector (Shimadzu Corp., Kyoto, Japan), as described by a previous report [32] with some modifications. A total of 0.05 g of cecal contents were acidified with 0.05 mL of sulfuric acid (FUJIFILM Wako Pure Chemical Corp., Osaka, Japan). Then, the SCFAs were extracted with a mixture of 0.4 mL of diethyl ether (FUJIFILM Wako Pure Chemical Corp., Osaka, Japan) and 0.2 mL of ethanol (FUJIFILM Wako Pure Chemical Corp., Osaka, Japan), and centrifuged at 18,700× *g* for 30 s. A total of 1 μL of the organic phase was injected into a capillary column (InertCap Pure WAX (30 m × 0.25 mm, df = 0.5 um) GL Science Inc., Tokyo, Japan) with an initial temperature of 80 °C and final temperature of 200 °C; helium was used as the carrier gas.

### 2.7. DNA Extraction from Feces

Fecal DNA extraction was performed as previously described, with modifications [33]. About 0.2 g of the fecal sample was mixed with 20 mL of PBS (Phosphate-buffered saline) and filtered through a 100 μm mesh nylon filter (Corning Inc., Corning, NY, USA). The debris on the filter was washed with 10 mL of PBS, and the filtrates were centrifuged at 9000× *g* for 20 min at 4 °C. Each precipitate was suspended with 1.5 mL of TE10 buffer (10 mM Tris-HCl, FUJIFILM Wako Pure Chemical Corp., Osaka, Japan/10 mM EDTA (Ethylenediamine tetraacetic acid), DOJINDO, Tokyo, Japan), centrifuged at 9560× *g* for 5 min at 4 °C and then suspended with 0.8 mL TE10 buffer. DNA was extracted using 1 mL PCI (Invitrogen, Carlsbad, CA, USA), and isolated using 0.1 mL lysozyme (FUJIFILM Wako Pure Chemical Corp., Osaka, Japan) and 0.02 mL achromopeptidase (FUJIFILM Wako Pure Chemical Corp., Osaka, Japan). DNA was purified using RNase (Promega Corp., WI, USA) and precipitated with 20% PEG (Polyethylene glycol) solution (Tokyo Chemical Industry Co., Ltd., Tokyo, Japan). It was rinsed with 70% ethanol and dissolved in 50 μL TE (Tris-EDTA) buffer.

### 2.8. 16 S rDNA Gene Sequencing

The 16S rDNA gene sequencing was performed according to the instructions of Illumina Inc. (San Diego, CA, USA). V3-V4 variable regions of the 16S rDNA gene were amplified by PCR using the following primers: Forward Primer = 5’-TCGTCGGCAGCGTCAGATGTGTATAAGAGACAGCCTACGGGNGGCWGCAG-3’, Reverse Primer = 5’-GTCTCGTGGGCTCGGAGATGTGTATAAGAGACAGGACTACHVGGGTATCTAATCC-3’.

PCR amplification was carried out using 2.5 μL microbial DNA (5 ng/μL), 5 μL of each primer (1 μmol/L), and 12.5 μL 2× KAPA HiFi HotStart Ready Mix (Kapa Biosystems Inc., Wilmington, MA, USA). It was performed at the following settings: 95 °C for 3 min, followed by 25 cycles at 95 °C for 30 s, 55 °C for 30 s, and 72 °C for 30 s. Finally, an extension was carried out at 72 °C for 5 min. The Amplicon PCR products were purified using AMPure XP beads (Beckman Coulter Inc., Brea, CA, USA) according to the manufacturer’s instructions. A Nextera XT Index kit v. 2 (Illumina Inc., San Diego, CA, USA) was used for the Illumina sequencing adapters and attachment of the dual indices. An index PCR was carried out using 5.0 μL PCR production, 5.0 μL Nextera XT Index Primer, 25 μL 2× KAPA HiFi HotStart Ready Mix, and 10 μL PCR-grade water. PCR was performed at the following settings: 95 °C for 3 min, followed by 8 cycles at 95 °C for 30 s, 55 °C for 30 s, and 72 °C for 30 s. Finally, an extension was carried out at 72 °C for 5 min. The index PCR products were purified using the AMPure XP beads (Beckman Coulter Inc., Brea, CA, USA). The quality of the purifications was evaluated using Agilent 2100 Bioanalyzer with a DNA1000 kit (Agilent Technologies Inc., Santa Clara, CA, USA). Then, the DNA library was diluted to 4 nmol/L.

The DNA library was sequenced using the Miseq reagent kit v. 3 (Illumina Inc., San Diego, CA, USA) in the Illumina Miseq 2× 300 bp platform. This sequencing was performed according to the manufacturer’s instructions.

### 2.9. Analysis of 16S rDNA Gene Sequences

Then, 16S rDNA sequencing reads were processed by the Quantitative Insights into Microbial Ecology (QIIME) pipeline v. 1.9.1 [34]. Then, the reads were assigned to operational taxonomic units (OTUs) at 97% identity with the UCLUST algorithm [35] and then compared to reference sequences in the Greengenes database (v. August 2013). A total of 3,197,966 reads was obtained from 94 samples; on average, 34,020 ± 2666 reads were obtained per sample. The taxonomy summary at the phylum and genus levels, the alpha diversity index (e.g., Simpson diversity), beta diversity index, and principal coordinate analysis (PCoA) were calculated using QIIME. A PCoA analysis was also calculated using unweighted UniFrac distances.

### 2.10. Statistical Analysis

The D’Agostino–Pearson test/Kolmogorov–Smirnov test and Bartlett’s test analyzed the data for normal or non-normal distributions, and equal or biased variations, respectively. If the data showed a normal distribution and equal variation, statistical significance was determined using one-way analysis of variance with a Tukey Test for post-hoc analysis. If the data showed a non-normal distribution or biased variation, statistical significance was determined using the Kruskal–Wallis test with Dunn’s Multiple Comparison Test for post-hoc analysis. A permutational multivariate analysis of variance (PERMANOVA) was performed by QIIME and used to assess changes in composition of the microbiota. All data are expressed as means ± standard error of the mean (SEM); *p* < 0.05 was considered statistically significant. Statistical analyses were performed using the GraphPad Prism v. 6.03 (GraphPad Software Inc., San Diego, CA, USA).

## 3. Results

### 3.1. Aim 1: Consumption of Jerusalem Artichokes Led to Greater Changes in Microbiota Composition than Inulin

In a preliminary experiment, we discovered that a HFD changed cecal pH and SCFA content when compared to a standard (EF) diet (Oriental Yeast Co. Ltd., Tokyo, Japan). Specifically, cecal pH (Appendix A) and levels of SCFAs (Appendix A) were found to be significantly higher and lower, respectively, in the HFD group than the EF diet group. In light of these findings, it was plausible that feeding mice a HFD may be an effective way to induce abnormal conditions for their microbiota.

The amount of fructan in inulin and Jerusalem artichokes was found to be nearly 100% and 40%, respectively (Figure 2A). No significant differences in food intake or change in body weight over the 10 days were observed among the groups (Figure 2B,C). Cecal pH was found to be significantly lower in the inulin and Jerusalem artichoke groups than the cellulose group (Figure 2D). Levels of acetic acid were higher for the inulin group when compared to the cellulose group (Figure 2E). Levels of lactic acid were significantly higher for the Jerusalem artichoke group when compared to the cellulose group (Figure 2G). Levels of propionic and butyric acid were significantly higher in the inulin and Jerusalem artichoke groups when compared to the cellulose group (Figure 2F,H).

Analyses revealed that the Simpson diversity index was not significantly different between the inulin and Jerusalem artichoke groups (Figure 2I). The beta diversity index was significantly different between the cellulose and inulin groups, the cellulose and Jerusalem artichoke groups, and the inulin and Jerusalem artichoke groups (Figure 3B–D, respectively).

The relative abundance of Actinobacteria was significantly higher in the inulin group than the cellulose group (Figure 4A). The relative abundances of *Lactococcus* and *Oscillospira* were significantly lower in the inulin and Jerusalem artichoke groups compared to the cellulose group (Figure 4I,K).

*Bifidobacterium*, *Parabacteroides*, *Streptococcus,* and *Allobaculum* experienced greater changes in the inulin group than the cellulose group (Figure 4D,F,J, L). *Prevotella* and *Lactobacillus* experienced more changes in the Jerusalem artichoke group than the cellulose group (Figure 4G,H).

### 3.2. Aim 2: Consumption of Both the Water-Soluble and Insoluble Components in Jerusalem Artichokes Changed the Composition of the Microbiota

We thought that the insoluble dietary fiber of Jerusalem artichoke might explain the reason why Jerusalem artichoke showed a different microbiota composition from inulin. Therefore, Jerusalem artichoke was dissolved in water and separated into a water layer and a precipitate layer by centrifugation; then, each layer was powdered by freeze-drying and given to mice for 10 days (Figure 1A).

The water-soluble components in the Jerusalem artichoke were found to consist of almost 60% fructan (Figure 5A). No significant differences in food intake or change in body weight over the 10 days were observed among the five groups (Figure 5B,C). Cecal pH was significantly lower in groups A and B + C than the control group (Figure 5D). Levels of acetic acid were significantly higher in group B than the control group and group C (Figure 5E). Levels of propionic and lactic acid were significantly higher in groups A and B + C than the control group (Figure 5F,G). Levels of butyric acid were significantly higher in groups A, B, and B + C than the control group (Figure 5H).

Analyses revealed that the Simpson index was significantly higher in group A than the control group and group C, as well as higher in group B + C than group C (Figure 5I). Per the beta diversity indices, the microbiota composition between group A and B + C group was significantly different, as well as the microbiota composition for the control group and group C against groups A, B, and B + C (Figure 6A,B, and Appendix A).

The relative abundance of Actinobacteria was significantly higher in groups A and B than the control group (Figure 7A). The relative abundance of Bacteroidetes was significantly higher in group B + C than group B (Figure 7B). The relative abundance of *Streptococcus* was significantly lower in groups A, B, and B + C than the control group (Figure 7J). The relative abundance of *Parabacteroides* was significantly lower in groups B and B + C than the control group (Figure 7F).

Shifts in levels of *Bifidobacterium*, *Bacteroides*, and *Oscillospira* were greater in groups A, B, and B + C than the control group (Figure 7D, E, and K, respectively).

### 3.3. Aim 3: The Organic Component in Jerusalem Artichokes Changed the Composition of the Microbiota

In the preceding aim, group B + C was fed half the amount of fructans as group B; however, both groups exhibited similar changes in microbiota composition. On the other hand, the insoluble group had no effect on the microbiota. In other words, it was possible that the insoluble component provided a strong effect when combination with the soluble component. Therefore, we extracted the organic soluble component using a chloroform/methanol vehicle to find out the role of organic components of Jerusalem artichoke on microbiota (Figure 1B).

No significant differences in food intake or change in body weight over the 10 days were observed among the seven groups (Figure 8A and B). Cecal pH was significantly lower in groups A and B’+D than the control group and group E (Figure 8C). Levels of lactic acid were significantly higher in groups A and B’ + D than the control group (Figure 8F). Levels of butyric acid were significantly higher in group A than the control group and groups D and E, as well as significantly higher in group B’ + D than the control group (Figure 8G).

No significant differences in the Simpson diversity index were detected among the groups (Figure 8H). Summarizing the beta diversity indices, the control group and groups A, E, and B’ + E exhibited independency from each other, as well as showed significantly different microbiota components than the other groups. Groups B’, D, and B’ + D displayed a similar microbiota composition, although each was significantly different from the other groups (Figure 9A,B, and Appendix A).

The relative abundance of *Bifidobacterium* was higher in group A than group E (Figure 10D). The relative abundance of *Allobaculum* was significantly higher in group A than group B’ + D (Figure 10L). The relative abundance of *Oscillospira* changed significantly for all groups except group E (Figure 10K).

## 4. Discussion

In the present study, the consumption of Jerusalem artichokes changed gut microbiota, increased concentrations of SCFAs, and decreased cecal pH in mice fed a HFD. When compared to inulin, interestingly, they were found to differentially impact the bacterial populations in the gut, but they had a similar influence on SCFA content and cecal pH. Reasons for this may be attributed to other bioactive compounds in Jerusalem artichokes known to modulate the microbiota, which is why we chose to fractionate the Jerusalem artichoke into water-soluble, insoluble, and organic-soluble extracts. From this procedure, we discovered the organic extract was primarily responsible for driving the changes in the microbiota composition, and that the organic and water-soluble extracts together boosted the production of SCFAs and lowered cecal pH. On the whole, it can be interpreted that these components play a stronger role in the change of the microbiota than inulin alone.

From the preliminary experiment, it was found that the consumption of Jerusalem artichokes and their individual components increased levels of SCFAs relative to a standard diet. However, the effect of these variables on microbiota composition has not yet been explored. In an earlier study comparing a standard diet, HFD, and HFD + inulin, the results showed each had a differential impact on microbiota composition [36]. In other words, a diet with inulin changed the composition of microbiota in a different manner than that of a standard diet. Given that Jerusalem artichokes contain a fair amount of inulin, we expect the intake of a HFD with Jerusalem artichokes changes to the different microbiota composition from standard diet.

Jerusalem artichoke powder was fractionated into soluble and insoluble components and then combined as food for the mice. Findings revealed that cecal pH decreased and SCFA content increased. However, the microbiota composition from group B + C was different from that of group A. This outcome may be on account of the ratio of water-soluble to insoluble extracts. The collection rates for the water-soluble and insoluble extracts were 81.9% and 18.1%, respectively, and in this study, they were combined at a ratio of 1:1 and fed to the mice. In previous studies, 1:1 and 3:1 ratios of functional oligosaccharides and insoluble dietary fiber resulted in differences in microbiota compositions [37]. For future experiments, we expect a water-soluble to insoluble extract ratio of 8:2 to have a similar effect to that of Jerusalem artichokes.

Inulin is known to increase the relative abundance of *Bifidobacterium* [38,39,40], and in our study, *Bifidobacterium* was increased upon the consumption of Jerusalem artichokes. This bacterium is known to produce acetic, lactic, and butyric acids [41,42], and we also saw levels lactic and butyric acids increase significantly. It has been reported that feeding yogurt supplemented with *Bifidobacterium breve* to infants helps prevent gastrointestinal disorders [43], so feeding Jerusalem artichokes to an individual may be equally beneficial in this regard.

*Oscillospira* is a butyric acid-producing bacterium [44] and studies in humans have uncovered a negative correlation between *Oscillospira* concentrations and body mass index, indicating that its relative abundance is reduced in obese individuals [45]. Moreover, higher amounts of this bacterium have been observed in the microbiota of constipated women over those who are healthy [46], suggesting that it increases when peristalsis is suppressed. In our study, *Oscillospira* was decreased upon consumption of Jerusalem artichokes. Therefore, intaking Jerusalem artichokes may improve constipation.

*Prevotella* is known to be relatively high in the digestive tracts of Asians, Latin Americans, and Africans. Its concentrations increase following intake of foods high in fiber, such as vegetables [47,48]. Here, *Prevotella* was specifically increased by feeding the Jerusalem artichoke, but not by any of its individual components or inulin alone. This finding underscores the importance of eating vegetables in their entirety for promoting gut health and may provide insight as to why the aforementioned population groups have higher levels of *Prevotella*.

Previous research has shown that α-diversity and relative abundances of *Lactobacillus* and *Bifidobacterium* are reduced in mice with dextran sodium sulfate (DSS)-induced colitis [49]. *Clostridium butyricum*, which produces butyric acid, is known to alleviate symptoms of DSS-induced colitis [50,51]. Although *Clostridium butyricum* was not detected in this study, levels of butyric acid, the relative abundance of *Bifidobacterium*, and α-diversity increased following intake of Jerusalem artichokes. Therefore, it is likely that Jerusalem artichokes and/or their individual components are capable of alleviating symptoms of colitis.

A limitation of this study was that we were unable to identify the exact organic components that affected the microbiota. The final collection rate of the organic extract was the lowest out of all the extracts, which is why quite a large amount of Jerusalem artichoke powder is required to identify organic extracts, which is costly and labor intensive. It is presumable that it consisted of lipids, such as essential oils, sesquiterpenes, and caffeoylquinic acids [52,53], based on current evidence. Additionally, previous research has noted that caffeoylquinic acids can increase *Bifidobacterium* levels [54], so it is possible that this essential oil, in particular, affected microbiota composition. Further investigations are warranted.

## 5. Conclusions

In summary, feeding Jerusalem artichokes changed the microbiota composition, increased SCFA production, and lowered cecal pH in mice fed a HFD. Components in the Jerusalem artichokes primarily responsible for this outcome, when consumed together, were the water-soluble extract (mainly inulin) and organic extract. This study provides evidence that powdered Jerusalem artichokes, rather than inulin supplements, may be superior for promoting SCFA production and preventing digestive diseases like colitis.

## Figures and Tables

**Figure 1 microorganisms-08-00954-f001:**
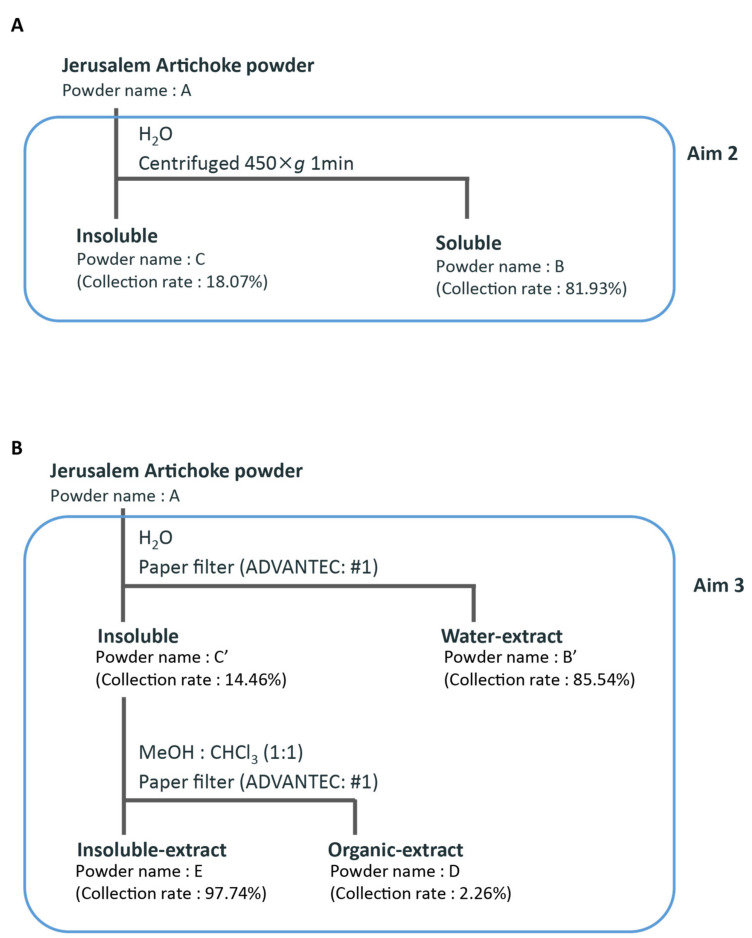
Fractionation scheme for (**A**) aim 2 and (**B**) aim 3.

**Figure 2 microorganisms-08-00954-f002:**
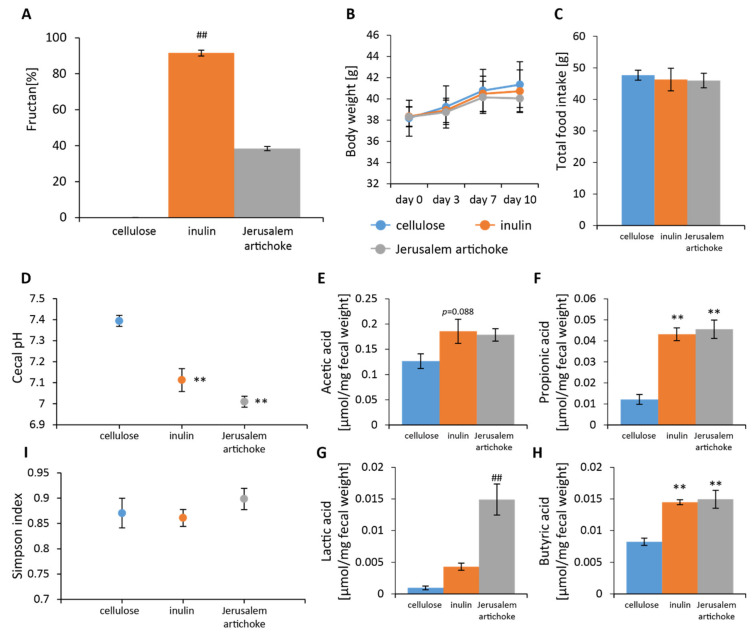
Consumptions of Jerusalem artichokes decreased cecal pH and increased levels of short-chain fatty acids similar to that of inulin. (**A**) The amount of fructan (cellulose (*n* = 4), inulin (*n* = 4), Jerusalem artichoke (*n* = 4)). (**B**) Change in body weight after 10 days. (**C**) Total food intake over 10 days. (**D**) Cecal pH of mice housed for 10 days for each group. (**E**–**H**) SCFAs of mice for 10 days, including (**E**) acetic acid, (**F**) propionic acid, (**G**) lactic acid, and (**H**) butyric acid. (**I**) Bacterial alpha diversity. Comparison of the Simpson diversity index for the 16S rDNA gene libraries at 97% similarity, per the sequencing analysis. All values are represented as mean ± SEM (cellulose (*n* = 9), inulin (*n* = 9), Jerusalem artichoke (*n* = 9)). ** *p* < 0.01 versus cellulose, evaluated using one-way ANOVA with Tukey post-hoc test. ## *p* < 0.01 vs. cellulose, evaluated using the Kruskal–Wallis with Dunn post-hoc test.

**Figure 3 microorganisms-08-00954-f003:**
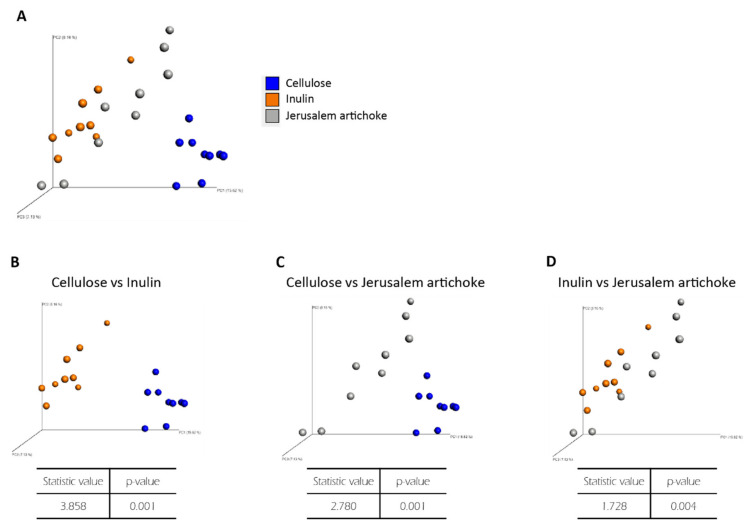
The microbiota compositions differed across all experimental groups. Bacterial beta diversity index for comparing (**A**) each group, (**B**) cellulose and inulin, (**C**) cellulose and Jerusalem artichoke, or (**D**) inulin and Jerusalem artichoke. The principal coordinate analysis (PCoA) plots of unweighted UniFrac distance metrics obtained from sequencing the 16S rDNA gene in feces (cellulose (*n* = 9), inulin (*n* = 9), Jerusalem artichoke (*n* = 9)). The table in these figures indicate the result using PERMANOVA.

**Figure 4 microorganisms-08-00954-f004:**
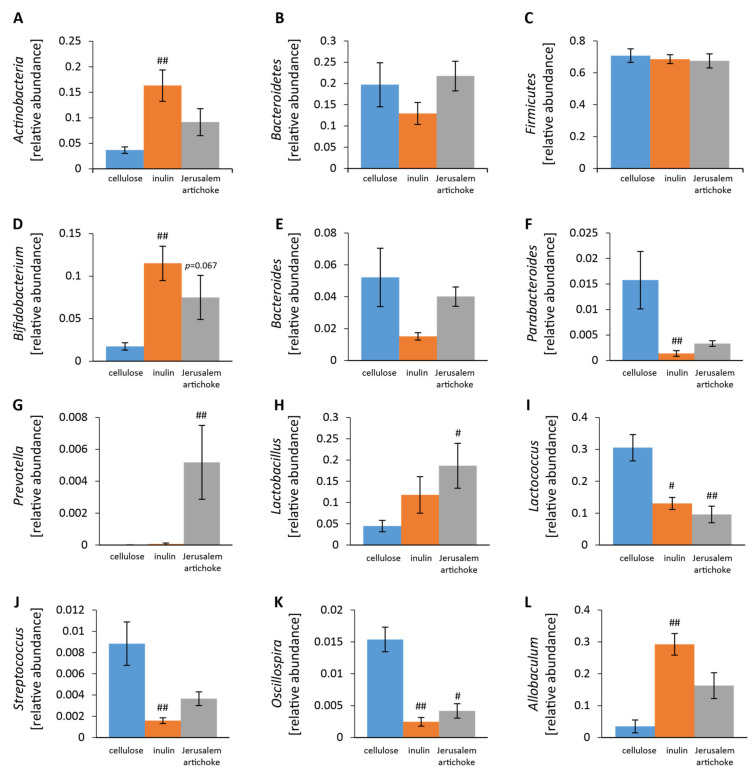
The relative abundance of microbes at the phylum and genus levels was changed upon consumption of inulin and the Jerusalem artichoke. (**A**–**C**) The relative abundance of microbes at the phylum level ((**A**) Actinobacteria, (**B**) Bacteroidetes, (**C**) Firmicutes). (**D**–**L**) The relative abundance of microbes at the genus level ((**D**) *Bifidobacterium*, (**E**) *Bacteroides*, (**F**) *Parabacteroides*, (**G**) *Prevotella*, (**H**) *Lactobacillus*, (**I**) *Lactococcus*, (**J**) *Streptococcus*, (**K**) *Oscillospira*, (**L**) *Allobaculum*). All values are represented as mean ± SEM (cellulose (*n* = 9), inulin (*n* = 9), Jerusalem artichoke (*n* = 9)). ## *p* < 0.01, # *p* < 0.05 vs. cellulose, evaluated using the Kruskal–Wallis with Dunn post-hoc test.

**Figure 5 microorganisms-08-00954-f005:**
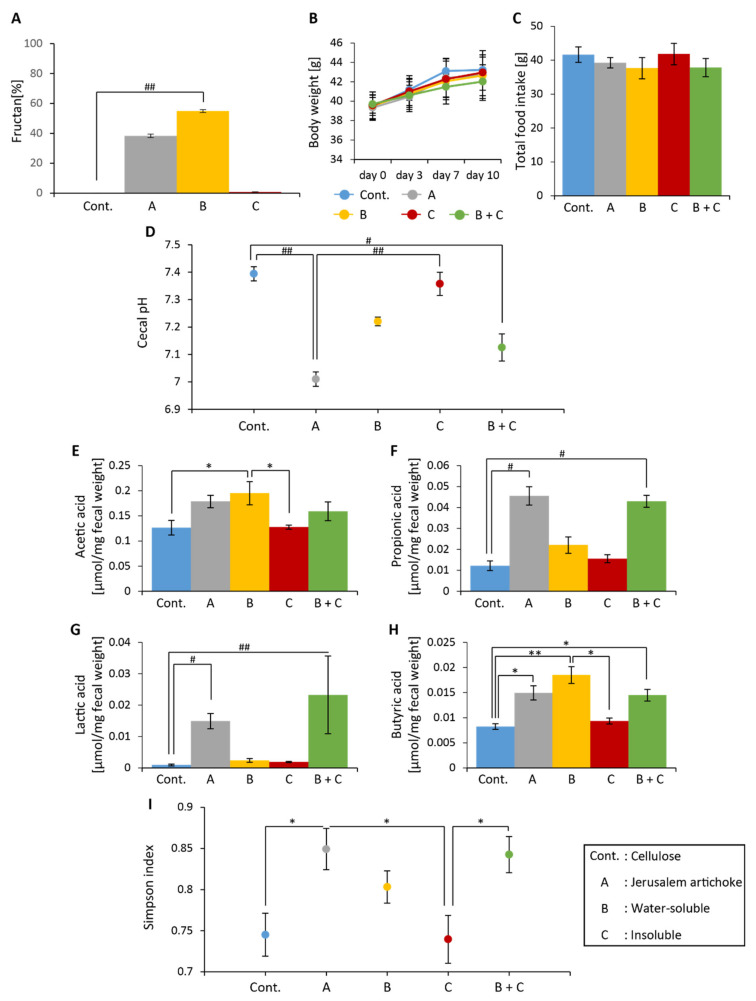
The Jerusalem artichoke, and the combination of water-soluble and insoluble extracts, decreased cecal pH, and increased levels of short-chain fatty acids. (**A**) The amount of fructan (cellulose (*n* = 4), Jerusalem artichoke (*n* = 4), water-soluble (*n* = 4), insoluble (*n* = 4)). (**B**) Change in body weight after 10 days. (**C**) Total food intake over 10 days. (**D**) Cecal pH of mice housed for 10 days for each group. (**E**–**H**) Short-chain fatty acids (SCFAs) of mice for 10 days, including (**E**) acetic acid, (**F**) propionic acid, (**G**) lactic acid, and (**H**) butyric acid. (**I**) Bacterial alpha diversity. Comparison of the Simpson diversity index for the 16S rDNA gene libraries at 97% similarity, per the sequencing analysis. All values are represented as mean ± SEM (cellulose (*n* = 5), Jerusalem artichoke (*n* = 5), soluble (*n* = 5), insoluble (*n* = 5), soluble + insoluble (*n* = 5)). ** *p* < 0.01, * *p* < 0.05, evaluated using One-way ANOVA with Tukey post-hoc test. ## *p* < 0.01, # *p* < 0.05, evaluated using the Kruskal–Wallis with Dunn post-hoc test. Cellulose, Jerusalem artichoke, water-soluble, or insoluble are Cont., A, B, or C, respectively.

**Figure 6 microorganisms-08-00954-f006:**
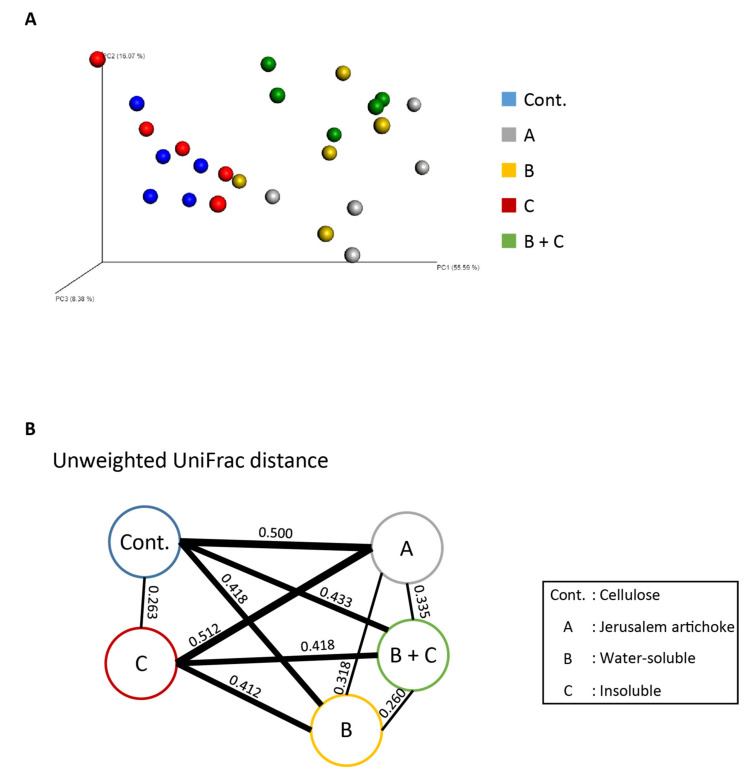
The Jerusalem artichoke and water-soluble extracts changed the microbiota composition. (**A**) Bacterial beta diversity for comparing each group. PCoA plots of unweighted UniFrac distance metrics obtained from sequencing the 16S rDNA in feces. (Cellulose (*n* = 5), Jerusalem artichoke (*n* = 5), soluble (*n* = 5), insoluble (*n* = 5), soluble + insoluble (*n* = 5)). (**B**) The average of unweighted UniFrac distance between groups. The number in (**B**) indicate the average distance calculated by UniFrac distance metrics. Cellulose, Jerusalem artichoke, soluble, or insoluble are Cont., A, B, or C, respectively.

**Figure 7 microorganisms-08-00954-f007:**
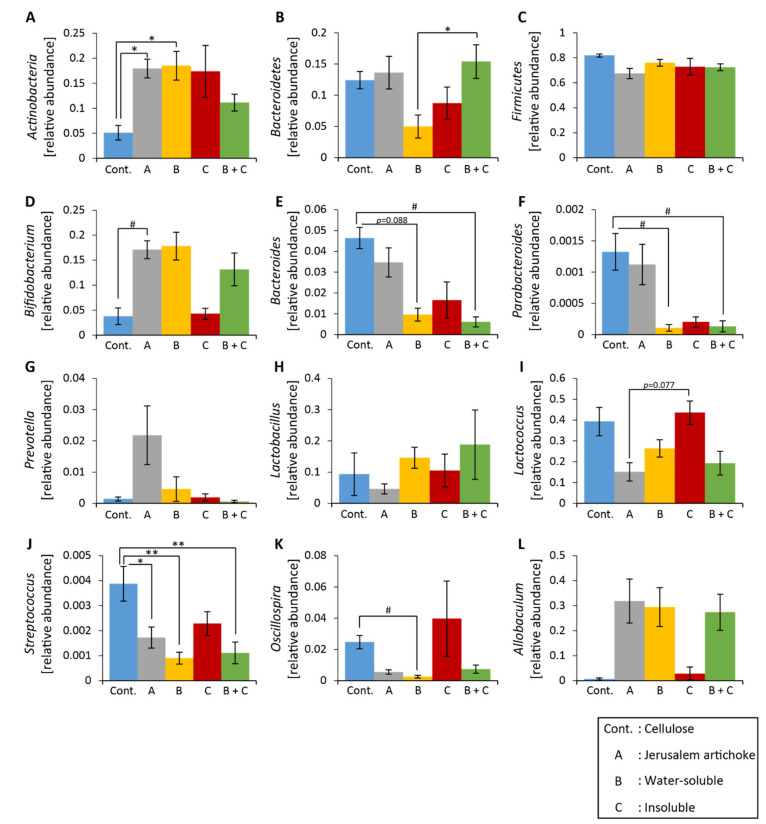
The relative abundance of microbes at the phylum and genus levels was changed upon consumption of the Jerusalem artichoke and the water-soluble extract. (**A**–**C**) The relative abundance of microbes at the phylum level ((**A**) Actinobacteria, (**B**) Bacteroidetes, (**C**) Firmicutes). (**D**–**L**) The relative abundance of microbes at the genus level ((**D**) *Bifidobacterium*, (**E**) *Bacteroides*, (**F**) *Parabacteroides*, (**G**) *Prevotella*, (**H**) *Lactobacillus*, (**I**) *Lactococcus*, (**J**) *Streptococcus*, (**K**) *Oscillospira*, and (**L**) *Allobaculum*). All values are represented as mean ± SEM (cellulose (*n* = 5), Jerusalem artichoke (*n* = 5), water-soluble (*n* = 5), insoluble (*n* = 5), water-soluble + insoluble (*n* = 5)). ** *p* < 0.01, * *p* < 0.05, evaluated using one-way ANOVA with Tukey post-hoc test. # *p* < 0.05, evaluated using the Kruskal–Wallis with Dunn post-hoc test. Cellulose, Jerusalem artichoke, water-soluble, or insoluble are Cont., A, B or C, respectively.

**Figure 8 microorganisms-08-00954-f008:**
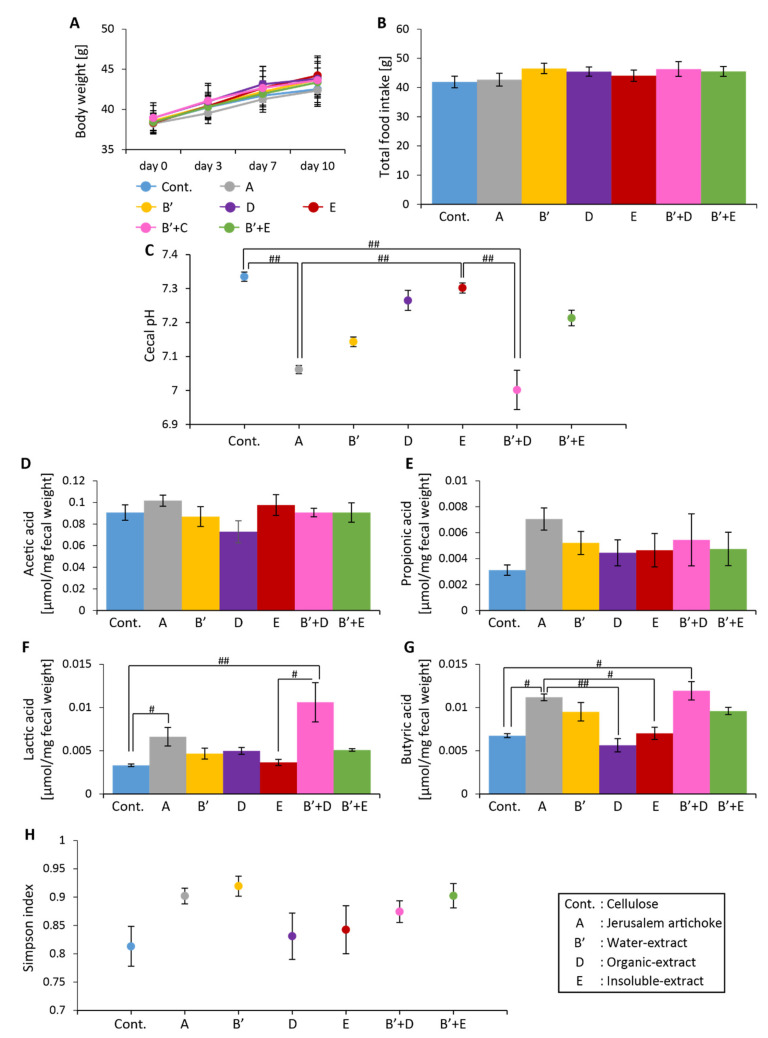
The Jerusalem artichoke, and the combination of the water-soluble and organic extracts, decreased cecal pH, and increased levels of short-chain fatty acids. (**A**) Change in body weight after 10 days. (**B**) Total food intake over 10 days. (**C**) Cecal pH of mice housed for 10 days for each group. (**D**–**G**) SCFAs of mice for 10 days, including (**D**) acetic acid, (**E**) propionic acid, (**F**) lactic acid, and (**G**) butyric acid. (**H**) Bacterial alpha diversity. Comparison of the Simpson diversity index for the 16S rDNA gene libraries at 97% similarity, per the sequencing analysis. All values are represented as mean ± SEM (cellulose (*n* = 6), Jerusalem artichoke (*n* = 6), water-soluble extract (*n* = 6), organic extract (*n* = 6), insoluble extract (*n* = 6), water-soluble extract + organic extract (*n* = 6), water-soluble extract + insoluble extract (*n* = 6)). ## *p* < 0.01, # *p* < 0.05, evaluated using the Kruskal–Wallis with Dunn post-hoc test. Cellulose, Jerusalem artichoke, water extract, organic extract, or insoluble extract are Cont., A, B’, D or E, respectively.

**Figure 9 microorganisms-08-00954-f009:**
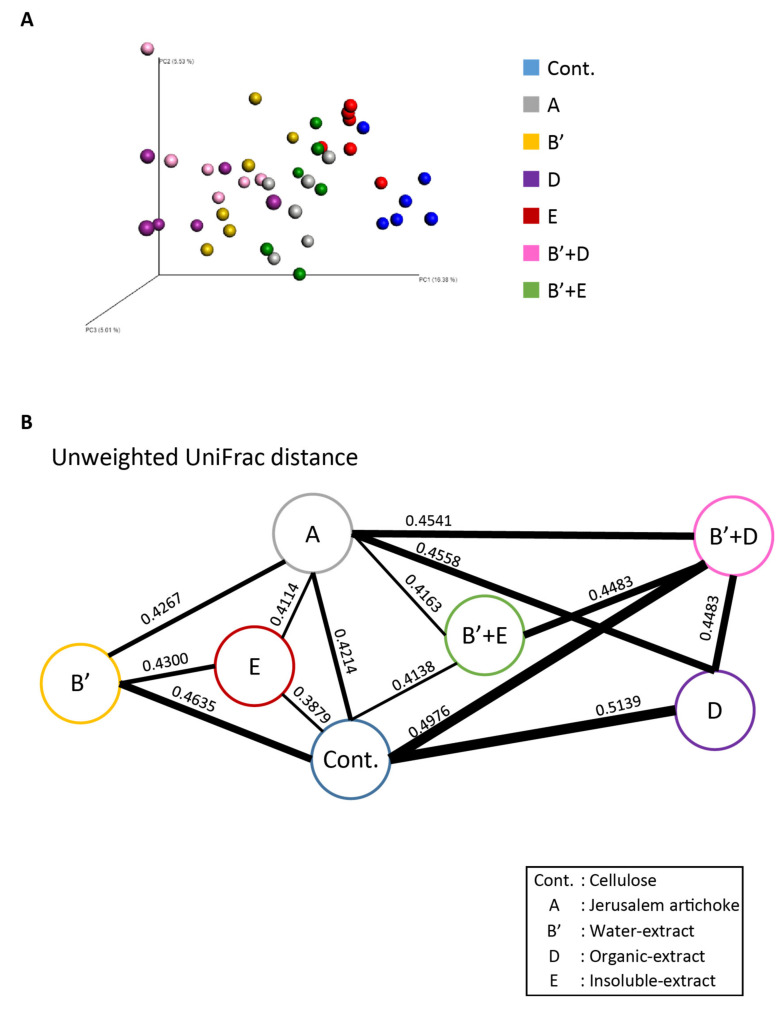
The water-soluble extract, organic extract component, and their combination resulted in similar change of the microbiota composition. (**A**) Bacterial beta diversity for comparing each group. PCoA plots of unweighted UniFrac distance metrics obtained from sequencing the 16S rDNA in feces. (Cellulose (*n* = 6), Jerusalem artichoke (*n* = 6), water-soluble extract (*n* = 6), organic extract (*n* = 6), insoluble extract (*n* = 6), water-soluble extract + organic extract (*n* = 6), water-soluble extract + insoluble extract (*n* = 6)). (**B**) The average of unweighted UniFrac distance between groups. The number in (**B**) indicates the average distance calculated by UniFrac distance metrics. Cellulose, Jerusalem artichoke, water-soluble extract, organic extract, or insoluble extract are Cont., A, B’, D, or E, respectively.

**Figure 10 microorganisms-08-00954-f010:**
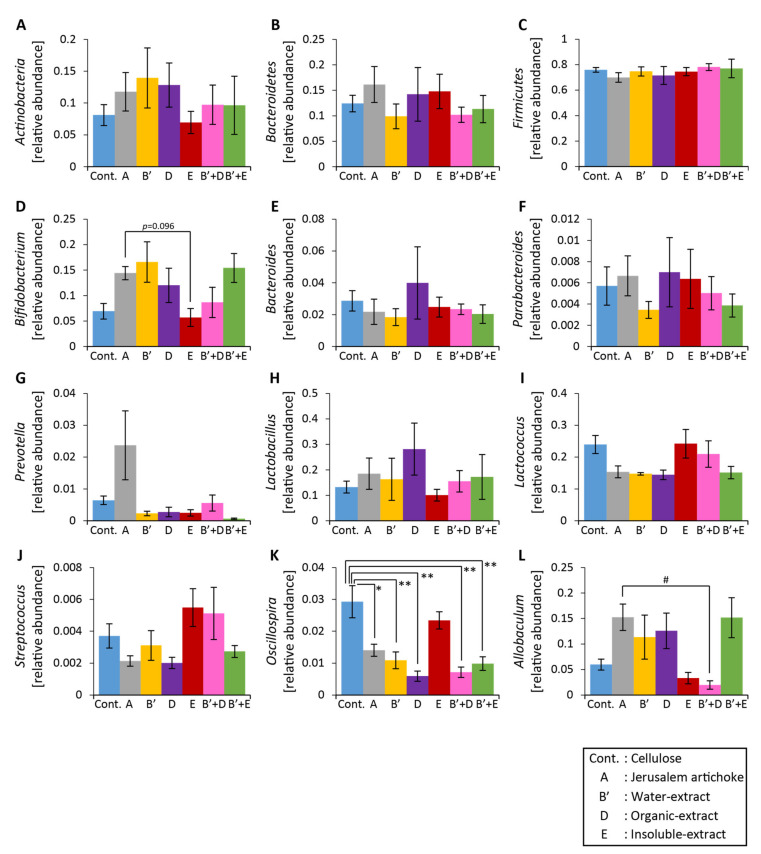
The relative abundance of microbes at the phylum and genus levels was hardly changed upon consumption of the Jerusalem artichoke, water-soluble extract, and organic extract. (**A**–**C**) The relative abundance of microbes at the phylum level ((**A**) Actinobacteria, (**B**) Bacteroidetes, (**C**) Firmicutes). (**D**–**L**) The relative abundance of microbes at the genus level ((**D**) *Bifidobacterium*, (**E**) *Bacteroides*, (**F**) *Parabacteroides*, (**G**) *Prevotella*, (**H**) *Lactobacillus*, (**I**) *Lactococcus*, (**J**) *Streptococcus*, (**K**) *Oscillospira*, (**L**) *Allobaculum*). All values are represented as mean ± SEM (Cellulose (*n* = 6), Jerusalem artichoke (*n* = 6), water-soluble extract (*n* = 6), organic extract (*n* = 6), insoluble extract (*n* = 6), water-soluble extract + organic extract (*n* = 6), water-soluble extract + insoluble extract (*n* = 6)). ** *p* < 0.01, * *p* < 0.05, evaluated using one-way ANOVA with Tukey post-hoc test. # *p* < 0.05, evaluated using the Kruskal–Wallis with Dunn post-hoc test. Cellulose, Jerusalem artichoke, water-soluble extract, organic extract, or insoluble extract are Cont., A, B’, D, or E, respectively.

**Table 1 microorganisms-08-00954-t001:** Nutritional information of the Jerusalem artichoke powder (10 g).

Energy	37 kcal
Protein	0.9 g
Fat	0.2 g
Carbohydrates	7.7 g
Salt	0.0 g

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
