# Peer review of "Combinatorial Effects of Soluble, Insoluble, and Organic Extracts from Jerusalem Artichokes on Gut Microbiota in Mice"

_microorganisms, 2020, doi:10.3390/microorganisms8060954_

Round 1

Reviewer 1 Report

It is an engaging article with well-designed approaches that purposefully question our knowledge of the effect of different fractions of Jerusalem artichokes extract on gut microbiota.

The methodology and results presented are not, however, in line with the manuscript title and the third purpose of the study defined as the investigation of the effect of "the bioactive compounds in the insoluble components of Jerusalem artichokes" since the bioactive components have not been identified at all. Addressing this issue is required since the Authors claim that they “discovered the organic extract was primarily responsible for driving the changes in the microbiota composition.”

Moreover, the presentation of some results, as listed below, is somewhat confusing.

  1. Line 285, the order of listed figures “Figures 7E, F and L, respectively,” is not in line with the order of microbes listed in line 284.
  2. Line 306 – “group D+E” is not presented in Figure 8G.
  3. Please also confirm that
  • “(…) the Soluble + Insoluble group contained half amount of fructans as compared to the Soluble group, Soluble + Insoluble and Soluble only groups” (lines 296-297)
  • the water-soluble extract decreased cecal pH and increased SCFA content (line 362)

It is also recommended to express the value of centrifugation speed in g-force instead of rpm (line 91).

Reviewer 2 Report

Dear Editor,

The paper by Sasaki et al, entitled « Combinatorial effects of soluble, insoluble and organic components from Jerusalem artichokes on gut microbiota in mice » presents the effects of different fractions obtained from artichokes on intestinal microbiota composition, on short chain fatty acids concentrations in the lumen and on the pH of the caecum. Interestingly, the authors identified that organic extracts from Jerusalem artichokes is necessary to decrease of the pH of the caecum and to the effects of Jerusalem artichokes on the microbiota composition. The paper is well written, the data are clearly presented and represent an advance in the field of prebiotics researches. However, I have questions regarding the experimental approach used by the authors:

Major comments

  • My main comment is that the control group used in the study is composed of high fat diet-fed mice receiving cellulose. A better control would be a group of mice fed a conventional diet, which would help to determine whether the different artichoke fractions limit the adverse effects of a high fat diet on intestinal microbiota composition. In the paper it is not clear which phenotype is beneficial or not. For example, the pH of caecum decreases in the artichoke group compared to the control group. However, what is the pH of a healthy caecum in mice fed a standard diet. For more clarity, I would suggest to add a group of mice fed a conventional diet in most of the experiments, when possible.

  • What is the effect of artichokes components on the microbiota of mice fed a conventional diet?

  • Which are the consequences of these modification on sensitivity of these mice to DDS-induced colitis? Are the mice receiving the different fractions of Jerusalem Artichokes protected against chemically-induced colitis?

  • The authors conclude by “powered Jerusalem artichokes, rather than inulin supplements, may be superior for promoting a healthy gut”. What do the authors call a “Healthy gut”? Is it a gut colonized by anti-inflammatory bacteria? Is it a gut colonized by a rich and diverse microbiota? Please precise and discuss this point.

Minor comments

  • page 4 line 114: What does ZT4 stands for? Please precise

  • page 6 line 195: “ Consumption of artichokes led to alterations in microbiota”. What do the author mean by alteration? Alteration generally refers to a detrimental phenotype. I would suggest “modifies” or “changes” instead of “alters”. The same comment applies to the other titles in the “Results” section. Of note, using a group of mice fed a conventional diet could help to determine if the artichokes-enriched diet “alters” or “beneficially modifies” the microbiota.

  • Figures 5-10: It is difficult to understand the graphs with the letters A, B, C… I suggest the authors add a frame on the figure to clearly describe what A, B C… refer to.

  • Figure 6A: Which groups were compared with the statistical test shown on the figure?

  • Figure 7: The legend of “Y” axis is missing on panel C. A,B and C are missing on the figure.

  • On the panel D of figure 7, the Proteobacteria profile seems very different from what is presented in figure 4D. The group A (Jerusalem Artichokes) shows the same relative quantity of Proteobacteria than the control group (Cellulose) whereas in figure 4A a slight difference is observed (p=0.067). Moreover, the scales between the two figures (4D and 7D) are very different. Could the authors explain these discrepancies?

Round 2

Reviewer 1 Report

  1. The title, referring to “organic components from Jerusalem artichokes,” is still not in line with the content of the manuscript, in which the effect of “organic extracts from Jerusalem artichokes” on gut microbiota in mice has been presented.
  2. Consequently, the information about the effect of the water-soluble extract on cecal pH and the concentration of SCFAs in the abstract (lines 23-24) should be corrected with the changes introduced in the discussion (lines 400-403).

Reviewer 2 Report

Dear Editor,

The authors properly answered my queries and modified the manuscript accordingly.

Of note, for an easier reading of the modifications made by the authors in the revised paper, I would have appreciated the authors indicate the line number in the point by point response to the reviewer instead of using “@@”.

Minor comment:

line 280 : « Ing » : What does that mean?

The following sentences are not clear to me:

line 398 : we expect intake of a HFD with Jerusalem artichokes to change to different pattern the different microbiota composition

line 400: When Jerusalem artichoke powder was first fractionated into soluble and insoluble components and then combined as food for the mice.
